# SKETCH: Semantic Key-Point Conditioning for Long-Horizon Vessel Trajectory Prediction

Linyong Gan [* 1]  Zimo Li [* 1]  Wenxin Xu [* 2]  Xingjian Li [* 1]  Jianhua Z. Huang [1 3]  Enmei Tu [4]  Shuhang Chen [4]

## Abstract

Accurate long-horizon vessel trajectory prediction remains challenging due to compounded uncertainty from complex navigation behaviors and environmental factors. Existing methods often struggle to maintain global directional consistency, leading to drifting or implausible trajectories when extrapolated over long time horizons. To address this issue, we propose a semantic-key-point-conditioned trajectory modeling framework, in which future trajectories are predicted by conditioning on a high-level Next Key Point (NKP) that captures navigational intent. This formulation decomposes long-horizon prediction into global semantic decision-making and local motion modeling, effectively restricting the support of future trajectories to semantically feasible subsets. To efficiently estimate the NKP prior from historical observations, we adopt a pretrain-finetune strategy. Extensive experiments on real-world AIS data demonstrate that the proposed method consistently outperforms state-of-the-art approaches, particularly for long travel durations, directional accuracy, and fine-grained trajectory prediction.

## 1. Introduction

Maritime transportation is the basis of global trade, carrying over 80% of goods worldwide by volume (Kosowska-Stamirowska, 2020). Accurate vessel trajectory prediction is therefore a fundamental capability for maritime intelligence, supporting collision avoidance (Papadimitrakis et al.,

2021; Zhang et al., 2022), port operation optimization (Peng et al., 2023), search and rescue (Liu et al., 2024), and fuel-efficient voyage planning (Zhang et al., 2025). Meanwhile, the widespread use of the Automatic Identification System (AIS) has enabled data-driven approaches by providing large-scale, high-frequency vessel motion records (Hexeberg et al., 2017; Murray & Perera, 2020; Wang et al., 2023b; Yang et al., 2022). However, the availability of large-scale data alone does not eliminate the intrinsic difficulty of long-horizon trajectory forecasting, where uncertainty compounds over time.

Building upon such data, existing approaches can be broadly categorized into non-learning and learning-based methods. Early non-learning approaches embed explicit mathematical models or statistical priors, including the Constant Velocity Model (CVM) (Xiao et al., 2020), the Nearly Constant Velocity (NCV) model (Ristic et al., 2008), Extended Kalman Filter (EKF) variants (Perera et al., 2012; Perera & Soares, 2010), stochastic processes such as the Ornstein–Uhlenbeck (OU) process (Millefiori et al., 2017), Gaussian Process (GP) models (Rong et al., 2019), and knowledge-based particle filtering methods (Mazzarella et al., 2015). Although computationally efficient and simple to implement (Chen et al., 2023; Xiao et al., 2020), these approaches often depend on simplified vessel-dynamics assumptions and neglect environmental effects (Kanazawa et al., 2021). Their robustness is therefore limited during maneuvers and in noisy scenarios (Chen et al., 2023; Gao et al., 2021), making them less suitable for medium- and long-horizon trajectory prediction (Nguyen & Fablet, 2024).

To overcome the limitations of purely statistical modeling, learning-based approaches have been widely explored for sequential trajectory prediction. Recurrent Neural Networks (RNNs) and Long Short-Term Memory (LSTM) networks improve model expressiveness but often struggle to capture long-range dependencies and multimodal future behaviors (Gao et al., 2021; Chen et al., 2023). In particular, MP-LSTM predicts midpoints and last points, and interpolates intermediate trajectories, which limits their ability to model fine-grained kinematic variations and directional changes (Gao et al., 2021). These limitations become increasingly pronounced as the prediction horizon grows.

---

[*]Equal contribution  [1]School of Data Science, The Chinese University of Hong Kong, Shenzhen, China [2]School of Science and Engineering, The Chinese University of Hong Kong, Shenzhen, China [3]School of Artificial Intelligence, The Chinese University of Hong Kong, Shenzhen, China [4]COSCO SHIPPING Advanced Technology Institute, Shanghai, China. Correspondence to: Enmei Tu <hellotem@hotmail.com>, Shuhang Chen <2480093@tongji.edu.cn>.

*Proceedings of the 43rd International Conference on Machine Learning*, Seoul, South Korea. PMLR 306, 2026. Copyright 2026 by the author(s).

More recently, Transformer-based models have shown promise due to their ability to model long-range dependencies via self-attention (Vaswani et al., 2017). Approaches such as TrAISformer enhance medium-term trajectory prediction and more effectively capture multimodal uncertainty (Nguyen & Fablet, 2024). Nevertheless, we observe that standard autoregressive Transformers still degrade severely in long-horizon settings. Notably, both TrAISformer (Nguyen & Fablet, 2024) and recent token-based GPT-style forecasters, such as TrackGPT (Stroh, 2024), convert continuous trajectory attributes into discrete tokens and learn embeddings over these discrete representations as the input to a Transformer backbone. Due to the discretization of continuous geographic space, token-based trajectory models are trained only on a sparse subset of spatial tokens. When deployed in previously unseen regions, the model must extrapolate beyond the empirical support of the training distribution, often leading to unstable or incoherent predictions. As uncertainty accumulates, predictions tend to collapse toward nearly horizontal trajectories. This observation suggests that increasing model capacity alone is insufficient for robust long-term trajectory forecasting.

One more limitation stems from the lack of explicit modeling of the global navigational intent. In real-world maritime navigation, vessels follow a hierarchical decision process: high-level route planning toward the NKPs, coupled with low-level continuous control of speed and heading (Potočnik, 2025). Most existing trajectory prediction models, however, operate solely at the local kinematic level. As a result, long-term navigational intent must be implicitly inferred from short-term motion patterns, which becomes increasingly unreliable over extended horizons.

Motivated by this observation, we recast long-horizon vessel trajectory prediction as a hierarchical forecasting problem that explicitly models global navigational intent. Our contributions are summarized as follows:

1. We identify the lack of explicit global intent modeling as a fundamental limitation of existing long-horizon vessel trajectory prediction methods.

2. We introduce the Next Key Point (NKP) as a semantic intent variable and condition it to trajectory prediction, separating global navigational decisions from local motion dynamics within a hierarchical framework.

3. We propose an efficient training strategy for NKP-conditioned forecasting, allowing the model to generalize to open-set navigational targets rather than depending on a fixed closed set of ports.

4. Experiments on large-scale AIS datasets show state-of-the-art performance, particularly in long-horizon prediction.

## 2. Related Works

### 2.1. Decoder-only Transformer Trajectory Predictor

Decoder-only Transformers, such as GPT-2 (Radford et al., 2019; Vaswani et al., 2017), consist of multiple blocks of masked multi-head self-attention and position-wise feed-forward layers, with residual connections, layer normalization, and dropout. They generate outputs autoregressively, predicting the next token conditioned on previous tokens:

$$p(x_{1:T}) = \prod_{t=1}^{T} p(x_t \mid x_{<t}). \tag{1}$$

Originally designed for natural language, decoder-only Transformers have also been successfully applied to sequential data in computer vision (Ramesh et al., 2021), audio generation (Wang et al., 2023a), and time series forecasting (Ansari et al., 2024). In our work, vessel trajectories are treated as sequences of tokens (latitude, longitude, SOG, COG), enabling autoregressive modeling of future positions within a vessel trajectory.

### 2.2. Retrieval-Augmented Verification

Retrieval-Augmented Detection (Kang et al., 2024) and Verification (Wang et al., 2025) reformulate prediction as similarity-based verification using reference samples from an external database. By matching query embeddings with stored embeddings, labels can be inferred beyond a fixed closed set, enabling open-set recognition and interpretable prediction.

We adapt this framework to NKP prediction by treating the Next Key Point as a semantic intent variable and retrieving stored trajectories whose hidden representations are most similar to the query:

$$\hat{h}_k = \arg\max_{h \in H} \mathrm{sim}_{\cos}(h_k, h) = \arg\max_{h \in H} \frac{h_k \cdot h}{\|h_k\| \cdot \|h\|}. \tag{2}$$

### 2.3. Contrastive Learning for Embedding Consistency

Contrastive learning (Hadsell et al., 2006) is used to ensure that trajectories leading to the same NKP have similar embeddings while different ones are pushed apart. For hidden states $H_1, H_2 \in \mathbb{R}^{B \times T \times H}$, average pooling and L2 normalization yield sequence-level embeddings $\hat{h}_1, \hat{h}_2$, and the cosine similarity $D_i = \mathrm{sim}_{\cos}(\hat{h}_1, \hat{h}_2)$ is computed. The loss function is:

$$\mathcal{L}_{\mathrm{TCL}} = \frac{1}{B} \sum_{i=1}^{B} \left[ (1 - y_i) D_i^2 + y_i \max(0, M - D_i)^2 \right], \tag{3}$$

where $y_i = 1$ if the pair shares the same NKP label, otherwise $y_i = 0$, and $M$ is a margin.

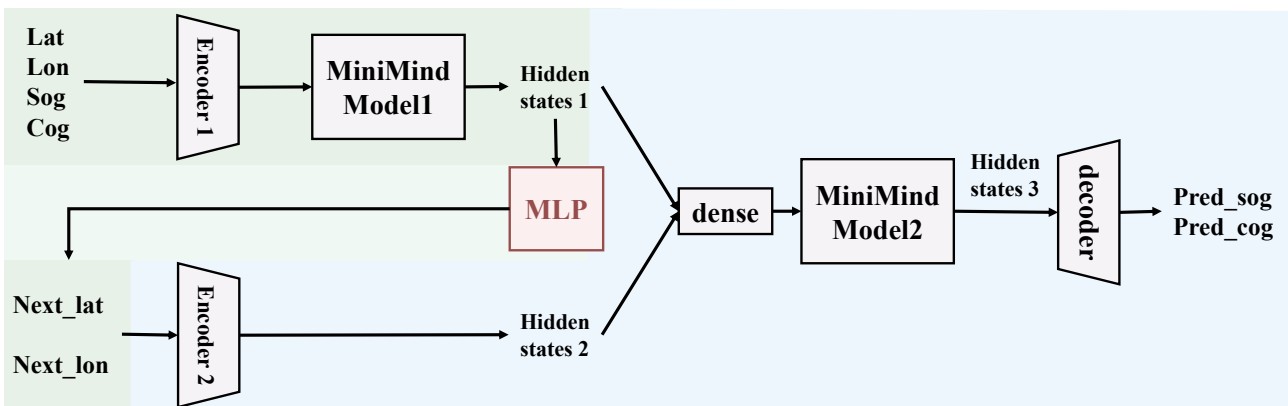

*Figure 1. Overall Architecture.* The inputs of AIS data go through Encoder 1 and MiniMind model 1 to be transformed into hidden state 1. The hidden state 1 will be sent into an MLP to predict the Next Key Point information. These coordinates will be sent to the encoder 2 to derive the hidden state 2. Then, hidden states 1 and 2 will be concatenated and passed through a dense layer, MiniMind model 2, and a decoder to obtain the predicted SOG and COG.

## 3. Methodology

### 3.1. Overview

To model vessel trajectories, we formulate the problem probabilistically. Let $X$ denote the observed historical trajectory, $Z$ the Next Key Point (NKP), and $Y$ the future trajectory. Our goal is to model the conditional distribution $p(Y \mid X)$, which we factorize as

$$P(Y \mid X) = \sum_Z P(Y \mid X, Z)\, P(Z \mid X), \qquad (4)$$

where $P(Z \mid X)$ captures the uncertainty of the Next Key Point, and $P(Y \mid X, Z)$ models the future trajectory given a specific NKP.

Conditioning on $Z$ refines future trajectory prediction by introducing high-level semantic constraints that restrict the support of admissible trajectories, i.e.,

$$\mathrm{supp}\big(P(Y \mid X, Z = z)\big) \subsetneq \mathrm{supp}\big(P(Y \mid X)\big). \quad (5)$$

This refinement should be understood as a reduction of predictive uncertainty rather than a pointwise increase of posterior likelihoods (see Appendix A).

NKP represents a semantic navigational constraint that defines a subset of feasible future trajectories rather than a specific spatial anchor. This factorization enables the model to disentangle high-level navigation decisions from low-level dynamics.

In addition, we formulate NKP inference in an open-set manner, allowing the model to generalize to newly emerging navigational targets. The concrete inference mechanism is detailed in Section 3.2, and the overall architecture is illustrated in Fig. 1.

### 3.2. Semantic Next-Key-Point Modeling

**Definition 3.1** (Semantic Next Key Point (NKP)). A Semantic Next Key Point (NKP) is an extendable predefined explicit coordinate point that represents a semantically meaningful maritime node which a vessel is likely to approach or pass in the near future, inferred from its observed trajectory history. Unlike an arbitrary geometric waypoint, an NKP belongs to a finite set of practically meaningful navigational anchors, such as port regions, straits, turning areas, or major shipping-lane junctions, and its semantic meaning lies in indicating the next important maritime node along the vessel's future movement.

NKPs are not the terminal points of trajectories, nor do they necessarily correspond to the final destination ports. Instead, they serve as semantically grounded segmentation points that decompose an extremely long trajectory into several acceptable shorter segments, each of which can be regarded as locally governed by a Markovian transition toward the next NKP, thereby enabling hierarchical long-horizon prediction.

**Remark.** We use the term *semantic* to emphasize that an NKP represents an intent-level equivalence class of feasible futures rather than a precise geometric waypoint, although in practice we obtain NKP supervision from spatial annotations (e.g., port/strait intersections) for training and evaluation.

To incorporate Next Key Point (NKP) information, we explicitly estimate the conditional distribution $p(Z \mid X)$, where $X$ denotes the observed trajectory history and $Z$ the corresponding NKP. Direct NKP classification scales poorly with dynamic maritime landscapes and cannot handle unseen key points, motivating a retrieval-based verification formulation. We adopt a contrastive verification-based formulation inspired by retrieval-augmented methods (Kang

**Algorithm 1** Next Key Point (NKP) Prediction

**Require:** Observed trajectory $X$
**Require:** Reference database $\mathcal{D} = \{(X_i, z_i)\}_{i=1}^{N}$
**Ensure:** Predicted Next Key Point $\hat{z}$
1: Compute embedding $h = f(X)$
2: **for** $(X_i, z_i) \in \mathcal{D}$ **do**
3:    Compute embedding $h_i = f(X_i)$
4:    $c(z_i) = \text{sim}(h, h_i)$
5: **end for**
6: $\hat{z} \leftarrow \arg\max_z c(z)$
7: **return** $\hat{z}$

et al., 2024), where destination inference is augmented by retrieving and verifying semantically similar trajectories from a reference database, which is well-suited for modeling semantic consistency among trajectories leading to the same destination.

As illustrated in Fig. 2, we construct training pairs consisting of trajectories associated with either the same or different NKPs. Each trajectory is encoded into a latent representation, and the cosine similarity between two representations is used to assess whether they correspond to the same NKP. Following the contrastive learning paradigm (Hadsell et al., 2006), we optimize the loss in Eq. (3), which encourages trajectories leading to the same NKP to share similar representations while separating those associated with different NKPs.

To improve efficiency, we freeze Encoder 1 and MiniMind Model 1 after pretraining and only fine-tune a lightweight multi-layer perceptron (MLP) for NKP verification. In this way, the learned representation acts as a compact semantic embedding, capturing high-level navigational intent. In practice, we approximate marginalization over $Z$ using a Maximum A Posteriori (MAP) estimate, which performs robustly in our experiments. When $P(Z \mid X)$ is sharply peaked, marginalization can be well approximated by conditioning on the MAP NKP.

During inference, an unknown trajectory is compared against a database of reference trajectories. The reference database contains only training trajectories for the private dataset. The largest similarity of each reference trajectory is cast for its prediction, as summarized in Algorithm 1. This procedure can be interpreted as a nonparametric approximation to $p(Z \mid X)$ via nearest-neighbor evidence aggregation. The overall training and inference pipeline is shown in Fig. 2.

### 3.3. Multi-stage Learning for NKP-Trajectory Modeling

This learning procedure provides a structured approximation to the joint distribution $p(Y, Z \mid X)$. As illustrated in Fig. 1, we adopt a three-stage framework that provides a structured approximation to the joint distribution $p(Y, Z \mid X)$, where $X$ denotes the observed trajectory history, $Z$ the Next Key Point (NKP), and $Y$ the future trajectory. The first two stages focus on learning the conditional components of the model, while the final stage performs integrated inference.

#### 3.3.1. CONDITIONAL TRAJECTORY MODELING (BLUE)

In the first stage, we train the trajectory prediction module to model the conditional distribution $p(Y \mid X, Z)$, where the ground-truth NKP is provided as oracle conditioning. This design decouples high-level navigational intent from low-level vessel dynamics, allowing the model to focus on long-horizon motion patterns under a fixed semantic constraint.

Although the Next Key Point $Z$ is unknown at inference time, we first learn $p(Y \mid X, Z)$ under oracle NKP supervision. This decouples high-level navigational intent from low-level motion dynamics, yielding a well-conditioned learning problem and preventing gradient interference when subsequently estimating $p(Z \mid X)$. To capture the sequential structure of vessel motion, the future trajectory distribution is factorized autoregressively as

$$p(Y \mid X, Z) = \prod_{t=1}^{T} p(y_t \mid X, Z, y_{<t}), \qquad (6)$$

and optimized using maximum likelihood with teacher forcing.

While autoregressive long-horizon prediction is expressive, it is susceptible to error accumulation due to exposure bias, arising from the mismatch between training-time conditioning on ground-truth prefixes and inference-time conditioning on model predictions. To mitigate this issue, we adopt a step-by-step training scheme, which stabilizes learning and reduces cumulative error propagation. From a reinforcement learning perspective, this procedure is equivalent to behavior cloning, where the model learns to imitate expert actions, i.e., SOG and COG of the vessels, conditioned on historical states and NKP information. These two training schemes were conducted alternatively. The loss functions for these two approaches are:

$$\mathcal{L}_{GPT2} = \frac{1}{T} \sum_{t=1}^{T} (\hat{vel}(t) - vel(t))^2 \qquad (7)$$

$$\mathcal{L}_{BC} = \frac{1}{T} \sum_{t=1}^{T} (\hat{coord}(t) - coord(t))^2 \qquad (8)$$

$vel$ refers to the SOG component over latitude and longitude, while $coord$ refers to the latitude and longitude. The latter loss function can be interpreted as minimizing a one-step imitation loss with respect to the expert policy induced

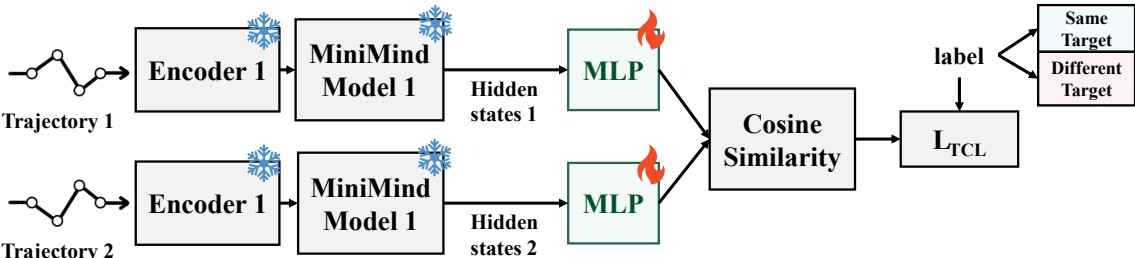

*Figure 2. Key-Point Prediction Training Paradigm.* Contrastive Learning is used to derive the hidden states of each trajectory, thereby decoupling the NKP information. To be more efficient, the two blocks trained previously are frozen and reused for fine-tuning.

by AIS trajectories. This model can be interpreted as the Maritime Trajectory Large Model for downstream tasks related to trajectory understanding.

### 3.3.2. NKP MODELING (GREEN)

In the second stage, we focus on estimating the NKP posterior $p(Z \mid X)$. The trajectory prediction backbone trained in Stage 1 is frozen, and only the NKP prediction feedforward module is optimized using the contrastive loss in Eq. (3). Freezing the backbone stabilizes representation learning and prevents interference between low-level motion dynamics and high-level semantic intent, ensuring that the learned NKP embeddings capture coarse navigational information rather than fine-grained kinematic details. With that, NKP can be derived as a byproduct.

### 3.3.3. INTEGRATED INFERENCE

During inference, the trained NKP module is first used to estimate the NKP $\hat{Z}$ from the observed trajectory history $X$. The future trajectory is then generated by conditioning the trajectory predictor on $(X, \hat{Z})$. This procedure provides a practical approximation to marginalizing over NKPs in the conditional distribution $p(Y \mid X)$, enabling coherent long-horizon trajectory prediction under NKP uncertainty.

### 3.4. Model Architecture and Physical Representation

To implement this factorization, we build our model upon a decoder-only Transformer (Vaswani et al., 2017), named MiniMind[1], due to its efficiency and strong temporal modeling capabilities. We introduce linear layers as input and output projections in the Transformer to process continuous maritime data, mapping latitude and longitude to continuous embeddings and SOG/COG to scaled action components. Specifically, we normalize the longitude and latitude to $[-1, 1]$ and convert COG and SOG into latitude/longitude velocity components using a scaling factor of $1/25$, which can be interpreted as the combined effect of crew decisions and environmental dynamics. However, to visualize the

---

[1] https://github.com/jingyaogong/minimind

motion, the next latitude and longitude should be calculated, given the predicted SOG and COG.

**Assumption 3.2** (Local Linear Motion). Over a short time interval $T$, the vessel is assumed to move with constant speed over ground (SOG) $v$ and course over ground (COG) $\theta$.

**Proposition 3.3** (SOG/COG Coordinate Update). *Under Theorem 3.2, given current coordinates $(lat_0, lon_0)$, the next position $(lat_1, lon_1)$ is given by*

$$lat_1 = lat_0 + \frac{v\cos\theta}{R}T, \tag{9}$$

$$lon_1 = lon_0 + \tan\theta \ln \frac{|\sec(lat_1) + \tan(lat_1)|}{|\sec(lat_0) + \tan(lat_0)|}, \tag{10}$$

*where $R$ denotes the Earth's radius.*

This formulation operates in a locally Euclidean space and avoids repeated spherical projections, thereby improving numerical stability and preventing systematic error amplification during long-horizon autoregressive rollout. The derivation follows from integrating the velocity field under a spherical-Earth approximation and is presented in Section C, along with a numerical stability analysis in Section D.

To empirically assess the numerical consistency of the proposed coordinate update, we perform a one-step displacement check on real AIS trajectories. For each trajectory segment of length 288, the next-step position is computed from the current SOG/COG and compared against the ground-truth next coordinate. Across the evaluated trajectories, the resulting mean squared error is on the order of $10^{-9}$, indicating numerical consistency up to machine precision.

## 4. Evaluation Metrics

**Mean Squared Error of Position (MSEP).** We evaluate point-level trajectory accuracy using the Mean Squared Error of Position (MSEP), defined as

$$\text{MSEP} = \frac{1}{T}\sum_{t=1}^{T}\left\|P^{\text{pred}}(t) - P^{\text{true}}(t)\right\|_2^2, \tag{11}$$

where $T$ denotes the prediction horizon, and $P^{\text{pred}}(t)$ and $P^{\text{true}}(t)$ represent the predicted and ground-truth positions at time step $t$, respectively.

MSEP measures the **point-wise positional discrepancy** between predicted and ground-truth trajectories and thus reflects local prediction accuracy. A smaller MSEP indicates more accurate trajectory predictions at the coordinate level.

**Mean Squared Error of Curvature (MSEC).** To evaluate trajectory smoothness rather than geometric curvature itself, we measure discrepancies in curvature profiles between predicted trajectories and the ground truth. We define the curvature at each trajectory point as

$$\kappa_i = \begin{cases} 0, & i \in \{0, n-1\}, \\ \frac{\Delta\theta_i}{\bar{d}_i}, & \text{otherwise,} \end{cases} \quad (12)$$

where $\Delta\theta_i$ denotes the change in heading angle at point $i$, and $\bar{d}_i$ is the average arc length of the adjacent segments. To reduce local noise, a smoothed curvature is computed using a three-point moving average:

$$\kappa_i^{\text{smooth}} = \frac{\kappa_{i-t} + \kappa_i + \kappa_{i+t}}{3}. \quad (13)$$

Based on the smoothed curvature, we define the Mean Squared Error of Curvature (MSEC) between the predicted trajectory and the ground truth as

$$\text{MSEC} = \frac{1}{n} \sum_{i=0}^{n-1} \left( \kappa_i^{\text{pred}} - \kappa_i^{\text{true}} \right)^2. \quad (14)$$

The MSEC evaluates trajectory quality at a microscopic level by measuring point-wise curvature discrepancies and thus reflects the **smoothness** of the predicted trajectory. The curvature definition follows the standard discrete approximation $\kappa \approx d\theta/ds$ commonly used in trajectory analysis and motion planning (LaValle, 2006).

**Mean Fréchet Distance (MFD).** The Mean Fréchet Distance (MFD) (Alt & Godau, 1995) is defined as

$$\text{MFD} = \frac{1}{B} \sum_{k=1}^{B} F(A_k, B_k), \quad (15)$$

where $F(\cdot, \cdot)$ denotes the discrete Fréchet distance between a predicted trajectory and its ground truth. This metric evaluates **curve-level** performance.

**Evaluation Metrics and Dataset Overview.** To comprehensively evaluate trajectory prediction performance at different granularities, we adopt three complementary metrics: Mean Squared Error of Position (MSEP), Mean Squared Error of Curvature (MSEC), Mean Fréchet Distance (MFD)

and inference duration. Specifically, MSEP measures point-wise positional accuracy and reflects local coordinate-level errors. MSEC evaluates discrepancies in curvature profiles and characterizes the smoothness and local geometric consistency of predicted trajectories. In contrast, MFD captures curve-level similarity by assessing the global geometric alignment between predicted and ground-truth trajectories. The inference durations, in seconds, evaluate the model's efficiency and include the time taken on the same test datasets at the same batch sizes. Together, these metrics provide a multi-scale evaluation framework that jointly accounts for local accuracy, trajectory smoothness, global shape consistency, and time efficiency. More datasets and implementation details were covered in Appendix Section E and Section F.

## 5. Results

### 5.1. Metrics on Trajectory Prediction

Table 2 reports a quantitative comparison between our method and several representative baselines, evaluated using three complementary metrics: MSEP, MSEC, and MFD.[2]

Overall, our method achieves the best performance across all reported metrics. Specifically, our model attains the lowest MSEP, indicating more accurate point-wise predictions over the entire time horizon. At the curve level, our approach also achieves the smallest MFD, reflecting closer geometric and curvature-level alignment with the ground-truth trajectories.

### 5.2. Generalization to Public AIS Dataset

We further evaluate the generalization capability of our method on the public AIS dataset (Xie et al., 2025), which is collected from a different source and exhibits a distribution distinct from our private dataset.

None of the evaluated models is trained or fine-tuned on this dataset, and all results therefore reflect true out-of-domain generalization performance. As shown in Table 3, our method achieves the best performance across all reported metrics, especially in the MFDs, which were considered the most critical metric. In contrast, TrAISformer exhibits severe performance degradation on this dataset, as its reliance on embedding layers prevents it from generalizing to unseen latitude-longitude trajectories.

### 5.3. Next Key Point Prediction Accuracy

This section evaluates the effectiveness of our Next Key Point (NKP) prediction models. We compare three variants: *pretrain-c*, the MiniMind model trained from scratch; *sft-c*,

---

[2]Implementation is available at https://github.com/LinyongGAN/SKETCH

*Table 1.* Qualitative comparison of MP-LSTM, TrAISformer, and our model. They are trained on the same training split with the same procedure, and evaluated on the same test split. Blue, green, red, orange, and purple trajectories refer to input, ground truth, our prediction, prediction from MP-LSTM (Gao et al., 2021), and prediction from TrAISformer (Nguyen & Fablet, 2024), respectively. Additional qualitative examples are provided in Appendix B.

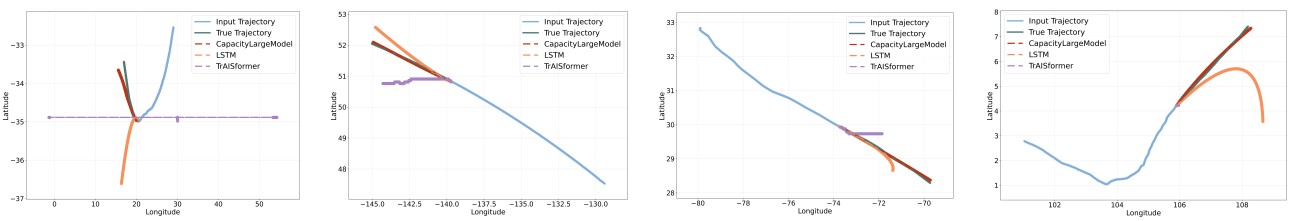

*Table 2.* Quantitative comparison of trajectory prediction performance between our method and other baselines, evaluated using MSEP, MSEC, and MFD on the whole test dataset.

| METRICS | MSEP | MSEC | MFD |
|---|---|---|---|
| OUR MODEL | **4.40E-5** | **1.24E-6** | **0.63** |
| MP-LSTM | 8.47E-4 | 1.6E-5 | 4.35 |
| TRAISFORMER | 0.35 | 0.015 | 12.95 |
| DIFFUTRAJ | 0.0011 | 1.6E-4 | 3.33 |
| AISFUSER | 0.25 | 0.0022 | 20.51 |

*Table 3.* Metrics for trajectory prediction on the public AIS_Dataset (Xie et al., 2025). All the metrics are the same as Table 2 with a total of 560 samples.

| METRICS | MSEP | MSEC | MFD |
|---|---|---|---|
| OUR MODEL | **2.81E-4** | **1.37E-5** | **0.97** |
| MP-LSTM | 3.83E-4 | 2.6E-5 | 2.77 |
| TRAISFORMER | 2.04 | 0.038 | 49.50 |
| DIFFUTRAJ | 7.9E-4 | 2.3E-4 | 2.91 |
| AISFUSER | 0.11 | 0.024 | 26.44 |

a closed-set probabilistic fine-tuned MLP; and *sft-o-s2*, an open-set supervised fine-tuned model capable of extending to unseen key points derived by contrastive learning.

*Table 4.* Next Key Point (NKP) prediction accuracy. Tested trajectories include NKPs absent from the sft-o-s2 database.

| METHOD | ACCURACY |
|---|---|
| SFT-O-S2 | 95.46% |
| SFT-C | 92.32% |
| PRETRAIN-C | **98.98%** |
| RANDOM FOREST(ZHANG ET AL., 2020) | 76.54% |

As shown in Table 4, all our proposed models significantly outperform the Random Forest baseline (Zhang et al., 2020). Among them, pretrain-c achieves the highest closed-set accuracy (98.98%), while sft-o-s2 also performs strongly (95.46%) and offers the advantage of seamless generalization to new NKPs through simple database extension, without retraining. The sft-c attains slightly lower accuracy

(92.32%). This demonstrates both the effectiveness of the fine-tuning procedure and its potential for application to other evolving prediction tasks.

Considering the trade-off between accuracy and flexibility, we adopt sft-o-s2 for integrated inference in subsequent experiments, as it provides robust performance while supporting scalable, open-set NKP prediction.

### 5.4. Effect of NKP Quality on Trajectory Prediction

We analyze the impact of Next Key Point (NKP) quality on the prediction of downstream trajectory in Table 5. The *correct* and *wrong* NKP settings define oracle upper and adversarial lower bounds, within which all models using predicted NKPs fall, indicating that NKP prediction errors induce only limited and acceptable degradation.

*Table 5.* Ablation of NKP Prediction. *Correct* and *wrong* NKP are oracle and adversarial inputs, while SFT-O, SFT-C, and pretrain-C are Stage 2 NKP prediction strategies. NKPs that do not appear in the database are omitted.

| METRICS | MSEP | MSEC | MFD |
|---|---|---|---|
| CORRECT NKP | **3.75E-5** | **1.21E-6** | **0.61** |
| SFT-O | 4.48E-5 | 1.69E-6 | 0.63 |
| SFT-C | 5.67E-5 | 1.93E-6 | 0.69 |
| WRONG NKP | 6.71E-4 | 4.45E-4 | 2.09 |
| PRETRAIN-C | 5.18E-3 | 3.20E-6 | 4.24 |
| PURE 4CH | 3.39E-4 | 7.25E-4 | 1.86 |

Among the learned NKP prediction strategies, *SFT-O* achieves the closest performance to the oracle *correct NKP* setting, with only slight degradation in MSEP, MSEC, and MFD. This indicates that open-set NKP prediction provides accurate and stable semantic guidance, motivating its use as the final NKP prediction scheme. In contrast, *SFT-C* performs worse across all metrics, suggesting that closed-set prediction is less robust in selecting appropriate future maritime anchors.

The *pretrain-C* model yields the largest MSEP and MFD despite its relatively low MSEC, showing that locally smooth

*Table 6.* Ablation Experiment for NKP Prediction. 4ch refers to the model without NKP information, 6ch refers to the model with correct NKP, while 6ch w/ WD refers to the model with random but wrong NKP.

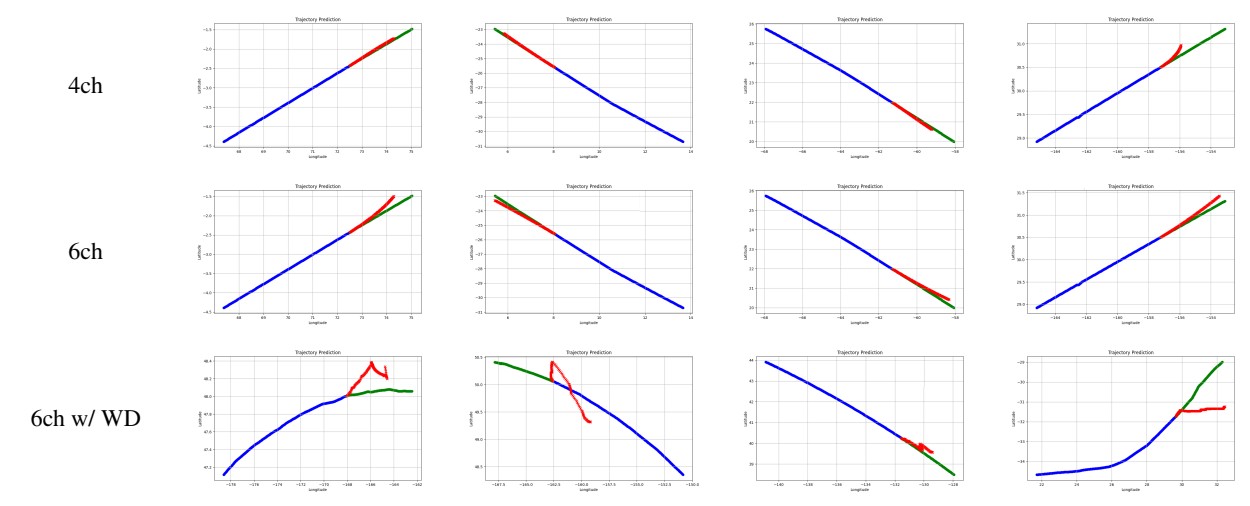

trajectories can still be globally inaccurate. This suggests that directly injecting non-temporal NKP information into temporal motion models, without explicitly decoupling NKP representation from motion modeling, may lead to ineffective or misleading semantic guidance.

Reliable NKP-conditioned models outperform the *pure 4ch* baseline, confirming that NKPs provide useful global intent beyond local motion cues. However, the *wrong NKP* setting causes substantial degradation and even performs worse than *pure 4ch* in MSEP and MFD, indicating that incorrect semantic guidance can distort global trajectory prediction. Overall, both NKP quality and integration are crucial for accurate and globally consistent long-horizon trajectory prediction.

### 5.5. Qualitative Results and Visualization

We qualitatively evaluate trajectory predictions against the ground truth across different regions (Table 1). MP-LSTM (Gao et al., 2021) predicts trajectories using only two anchor points (support and destination). While the support point is generally more accurate, the destination is often less reliable, and an error in either anchor can induce a significant global deviation in the reconstructed trajectory. TrAISformer (Nguyen & Fablet, 2024), applied from the official repository, shows pauses and stagnant latitude. In contrast, our model accurately captures both straight and turning trajectories, maintaining correct timing and coordinates, which demonstrates robust long-horizon prediction in complex scenarios.

Table 6 highlights the effectiveness of the proposed NKP channel. Models with NKP consistently produce trajectories closer to the ground truth than those without it, using the

same random seed for a fair comparison. The 6-channel row with an incorrect NKP input shows that the model effectively decouples NKP information: errors in NKP cause the predicted trajectory to twist, confirming the model's sensitivity to NKP guidance while retaining robustness.

### 5.6. Performance Across Prediction Horizons

Table 7 presents the MFD results over a time span ranging from 12 points to 144 points for the prediction length. As the prediction time range increases, the MFDs of all comparison methods show a consistent upward trend. However, our model achieves the best performance across all prediction time ranges, demonstrating its stability in short- and long-term trajectory prediction.

## 6. Conclusion

We presented a hierarchical framework for long-horizon vessel trajectory prediction that incorporates Next Key Points (NKPs) as explicit semantic intent. By decoupling high-level intent inference from conditional motion modeling, the framework reduces long-term uncertainty and improves trajectory coherence.

Beyond maritime navigation, this intent-conditioned factorization provides a general modeling paradigm for other long-horizon sequential prediction problems with hierarchical decision structures. Moreover, our approach can serve as a foundation for developing maritime AI solutions that can enable more effective trajectory understanding and offer a base representation for downstream tasks such as anomaly detection, route optimization, or multi-agent coordination.

*Table 7.* Horizon-wise. MFD for each model under different prediction horizon from 12 points to 144 points.

| MFD | 12 | 24 | 36 | 48 | 60 | 72 | 84 | 96 | 108 | 120 | 132 | 144 |
|---|---|---|---|---|---|---|---|---|---|---|---|---|
| OUR MODEL | **0.012** | **0.033** | **0.061** | **0.097** | **0.139** | **0.189** | **0.245** | **0.308** | **0.379** | **0.457** | **0.541** | **0.631** |
| MP-LSTM | 0.310 | 0.506 | 0.586 | 0.559 | 0.470 | 0.417 | 0.533 | 0.993 | 1.640 | 2.418 | 3.321 | 4.347 |
| TRAISFORMER | 5.090 | 5.721 | 6.630 | 7.507 | 8.129 | 8.672 | 9.351 | 10.077 | 10.977 | 11.568 | 12.352 | 12.949 |
| DIFFUTRAJ* | 1.980 | 2.169 | 2.260 | 2.347 | 2.419 | 2.518 | 2.633 | 2.737 | 2.857 | 2.992 | 3.132 | 3.327 |
| AISFUSER* | 2.009 | 3.731 | 5.353 | 6.749 | 7.926 | 9.578 | 10.816 | 12.502 | 13.805 | 16.702 | 18.943 | 20.509 |

* Closed-source; implemented according to its original paper.

## Impact Statement

This work studies long-horizon trajectory prediction for civil container vessels based on historical navigation data. Potential positive impacts include supporting strategic-level maritime transportation analysis, such as providing voyage-status estimates for ship operators and cargo owners, and enabling long-term route and schedule assessment under varying environmental conditions. These applications may improve operational planning and overall efficiency in maritime logistics. The data used in this study consist of historical vessel trajectories and do not involve personal or sensitive information.

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

# A. On Information Gain by Conditioning

In this appendix, we formally clarify the relationship between conditioning on additional information and the reduction of uncertainty. While conditioning on an auxiliary variable does not necessarily increase posterior probabilities pointwise, it always improves certainty in an information-theoretic or decision-theoretic sense.

## A.1. Preliminaries

Let $(\Omega, \mathcal{F}, \mathbb{P})$ be a probability space and let $X, Y, Z$ be random variables defined on it. Denote by $\sigma(X)$ and $\sigma(X, Y)$ the $\sigma$-algebras generated by $X$ and $(X, Y)$ respectively, with $\sigma(X) \subseteq \sigma(X, Y)$.

We use $H(\cdot)$ to denote Shannon entropy and $I(\cdot; \cdot \mid \cdot)$ to denote conditional mutual information.

## A.2. Conditioning and Uncertainty Reduction

We begin with a fundamental result stating that conditioning on additional information cannot increase uncertainty.

**Lemma A.1** (Monotonicity of Conditional Entropy). *For any random variables $X, Y, Z$, the following inequality holds:*

$$H(Z \mid X, Y) \ \leq \ H(Z \mid X). \tag{16}$$

*Proof.* By the chain rule of entropy, we have

$$H(Z \mid X) - H(Z \mid X, Y) = I(Z; Y \mid X), \tag{17}$$

where $I(Z; Y \mid X)$ denotes the conditional mutual information. Since mutual information is non-negative, i.e., $I(Z; Y \mid X) \geq 0$, the result follows. $\square$

Lemma A.1 provides a precise mathematical interpretation of the statement that *conditioning on additional information increases certainty*: the posterior uncertainty of $Z$ is reduced in an information-theoretic sense.

## A.3. Why Pointwise Posterior Improvement Is Not Guaranteed

Despite the reduction in entropy, conditioning on $Y$ does not generally imply a pointwise increase of posterior probabilities.

**Lemma A.2** (Tower Property of Conditional Expectation). *For any random variables $X, Y, Z$, we have*

$$\mathbb{E}[\mathbb{P}(Z \mid X, Y) \mid X] = \mathbb{P}(Z \mid X). \tag{18}$$

*Proof.* This follows directly from the tower property of conditional expectation:

$$\mathbb{E}[\mathbb{E}[\mathbf{1}_Z \mid X, Y] \mid X] = \mathbb{E}[\mathbf{1}_Z \mid X] = \mathbb{P}(Z \mid X). \tag{19}$$

$\square$

Lemma A.2 implies that any pointwise monotonic inequality such as $\mathbb{P}(Z \mid X, Y) \geq \mathbb{P}(Z \mid X)$ cannot hold almost surely unless equality holds everywhere. Therefore, the benefit of conditioning should not be interpreted as a pointwise increase in posterior probabilities.

## A.4. Decision-Theoretic Interpretation

The information gain induced by conditioning admits an equivalent interpretation in decision theory.

**Theorem A.3** (Bayes Risk Monotonicity). *Let $L$ be any loss function. Then*

$$\inf_{\hat{Z}(X, Y)} \mathbb{E}[L(Z, \hat{Z})] \ \leq \ \inf_{\hat{Z}(X)} \mathbb{E}[L(Z, \hat{Z})]. \tag{20}$$

*Proof.* Any decision rule based on $X$ alone is a special case of a decision rule based on $(X, Y)$ that ignores $Y$. Taking the infimum over a larger class of decision rules cannot increase the minimum achievable risk. $\square$

Theorem A.3 formalizes the intuition that additional information cannot worsen optimal decision performance.

### A.5. Remarks on Stronger Monotonicity Notions

We emphasize that stronger notions of monotonicity, such as pointwise improvement of posterior probabilities or monotonicity in likelihood ratio order, require additional structural assumptions on the conditional distribution $P(Y \mid Z, X)$. Without such assumptions, uncertainty reduction should be understood exclusively in an information-theoretic or decision-theoretic sense, as established above.

## B. Additional Qualitative Results

In this appendix, we provide additional qualitative comparisons between MP-LSTM, TrAISformer, and our method on diverse navigation scenarios.

*Table 8.* Additional qualitative trajectory prediction examples.

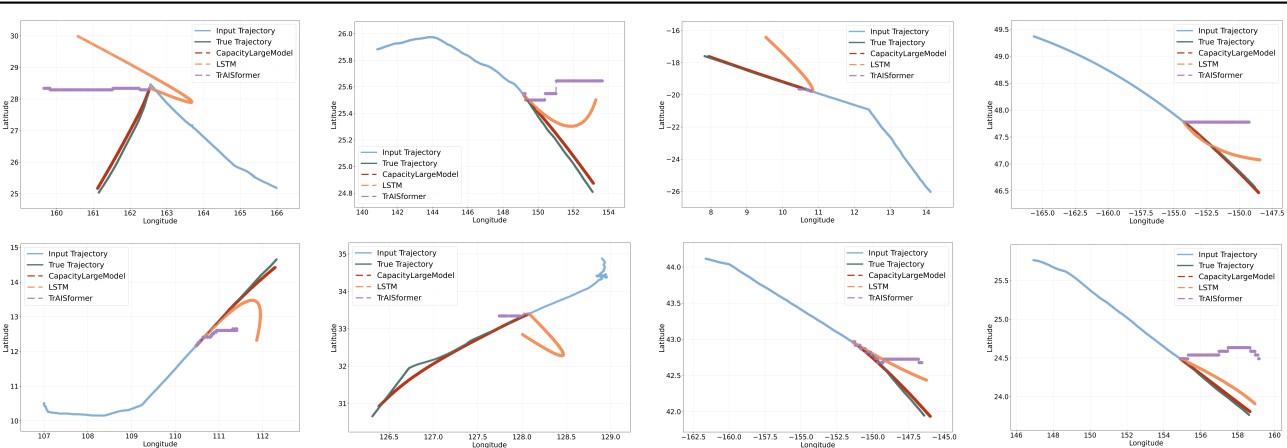

## C. Detailed Derivation of Next-Timestamp Coordinates

Given current coordinates $lat_0, lon_0$, current SOG $v$, and COG $\theta$, with a given short time period $T$, the new vessel coordinates $lat_1, lon_1$ were expected to be calculated. Assume that the vessels perform uniform linear motion in the short time period. Then the coordinates in the next timestamp are:

Then, in the latitudinal direction, the SOG components are:

$$v_{lat} = v \, \cos \theta \tag{21}$$

As the radii remain,

$$\omega_{lat} = \frac{v_{lat}}{R} \tag{22}$$

And the changes in latitude are,

$$\Delta lat = \omega_{lat} t = \frac{vt \cos \theta}{R} \tag{23}$$

The relationships between latitude and time are

$$lat(t) = lat_0 + \frac{v \cos \theta}{R} t \tag{24}$$

In the longitudinal direction, the SOG components are:

$$v_{lon} = v \sin \theta \tag{25}$$

The relationships between radii and latitude are

$$r = R\cos(lat) \tag{26}$$

The angular velocity is

$$\omega_{lon} = \frac{v\sin\theta}{R\cos(lat)} \tag{27}$$

Take (24) into (27) that

$$\omega_{lot}(t) = \frac{v\sin\theta}{R\cos(lat)} = \frac{v\sin\theta}{R\cos(lat_0 + \frac{v\cos\theta}{R}t)} \tag{28}$$

Then the changes of longitudes are

$$\Delta lon = \int_0^t \omega_{lon}(t)dt = \frac{v\sin\theta}{R}\int_0^t \frac{dt}{\cos(lat_0 + \frac{v\cos\theta}{R}t)} \tag{29}$$

With the formula

$$\int \frac{1}{\cos\theta}d\theta = \ln|\sec\theta + \tan\theta| + C \tag{30}$$

We can derive

$$\Delta lon = \frac{Rv\sin\theta}{v\cos\theta R}\int_0^t \frac{1}{\cos(lat_0 + \frac{v\cos\theta}{R}t)}d(\frac{v\cos\theta}{R}) \tag{31}$$

$$= \tan\theta(\ln|\sec\theta + \tan\theta|)\Big|_{lat_0}^{\frac{v\cos\theta}{R}T + lat_0} \tag{32}$$

Denoting $lat_1 = lat_0 + \frac{v\cos\theta}{R}T$

$$\Delta lon = \tan\theta\ln\frac{|\sec(lat_1) + \tan(lat_1)|}{|\sec(lat_0) + \tan(lat_0)|} \tag{33}$$

Finally, we can conclude that the new coordinates are

$$lat_1 = lat_0 + \frac{v\cos\theta}{R}T \tag{34}$$

$$lon_1 = lon_0 + \tan\theta\ln\frac{|\sec(lat_1) + \tan(lat_1)|}{|\sec(lat_0) + \tan(lat_0)|} \tag{35}$$

## D. Numerical Stability of Autoregressive Rollout

We analyze the numerical behavior of autoregressive trajectory rollout under different coordinate update formulations.

Let $s_t \in \mathbb{R}^d$ denote the true vessel state at time $t$ and $\hat{s}_t$ the predicted state, with prediction error $e_t = \hat{s}_t - s_t$. In autoregressive forecasting, predictions are propagated according to

$$\hat{s}_{t+1} = f(\hat{s}_t), \tag{36}$$

which induces the first-order error dynamics

$$e_{t+1} \approx J_f(s_t)\,e_t, \tag{37}$$

where $J_f$ denotes the Jacobian of the state transition function evaluated at $s_t$. Long-horizon numerical stability therefore depends critically on the spectral properties of $J_f$.

### D.1. Spherical Geometry–Based Updates

In spherical or geodesic formulations, state updates are typically implemented as a composition of coordinate projection and motion integration:

$$f_{\text{sph}} = \Pi^{-1} \circ G \circ \Pi, \tag{38}$$

where $\Pi$ maps geographic coordinates to a local tangent plane, and $G$ performs motion updates in that plane.

The Jacobians of $\Pi$ and $\Pi^{-1}$ depend nonlinearly on latitude and heading, introducing location-dependent scaling factors. The resulting error propagation becomes

$$e_{t+1} \approx J_{\Pi^{-1}} \, J_G \, J_{\Pi} \, e_t. \tag{39}$$

Even if $J_G$ is well-conditioned, repeated application of $J_{\Pi}$ and $J_{\Pi^{-1}}$ across time steps leads to multiplicative error accumulation. As a result, small numerical inaccuracies can be systematically amplified during long-horizon rollout, especially at high latitudes or under frequent heading changes.

### D.2. Locally Euclidean Update (Ours)

In contrast, our formulation performs state updates directly in a locally Euclidean coordinate system consistent with AIS-reported SOG and COG. Under the local linear motion assumption, the update function is smooth and approximately affine over short time intervals:

$$f_{\text{ours}}(s_t) = s_t + \Delta t \cdot v(s_t), \tag{40}$$

where $v(\cdot)$ denotes the velocity field induced by SOG and COG.

The corresponding Jacobian satisfies

$$J_f \approx I + \mathcal{O}(\Delta t), \tag{41}$$

where $I$ is the identity matrix. Consequently, prediction errors accumulate approximately additively rather than multiplicatively:

$$\|e_T\| \leq \|e_0\| + \mathcal{O}(T\Delta t). \tag{42}$$

This behavior contrasts sharply with spherical formulations, where the effective Jacobian deviates from identity due to repeated nonlinear projections. Therefore, our locally Euclidean update is numerically more stable under long-horizon autoregressive rollout and prevents systematic drift caused by geometric distortion.

## E. Dataset and Data Preprocessing

### E.1. Data Description and Coverage for Private Dataset

The dataset is proprietary and cannot be publicly released due to licensing restrictions. However, all baseline methods are trained and evaluated on the same dataset under identical preprocessing and evaluation protocols. Each data point includes the current timestamp, current longitude and latitude, current Speed Over Ground (SOG), Course Over Ground (COG), and the destination port name with its corresponding coordinates.

Our dataset contains comprehensive global maritime coverage with the following geographical boundaries:

- Latitude range: $36.4°S$ to $55.5°N$

- Longitude range: $180.0°W$ to $180.0°E$

This extensive spatial coverage includes major shipping routes across the Atlantic, Pacific, and Indian Oceans, as well as significant regional seas, ensuring the model's exposure to diverse maritime navigation patterns. All vessels in this private dataset are ocean-going container ships engaged in long-haul international trade.

### E.2. Data Preprocessing Pipeline

The data preprocessing consists of two steps:

**Step 1 - Temporal Interpolation:** Due to the non-uniform temporal distribution with AIS system limitation, we perform interpolation with a fixed time interval $T = 5$ minutes to achieve uniform sampling.

**Step 2 - Spatial Annotation:** The Next Key Point is determined by intersecting each trajectory with predefined maritime key nodes (e.g., ports and straits), and all points between two consecutive intersections are labeled with the corresponding NKP, yielding 103 unique spatial annotations.

After uniform interpolation, the dataset comprises 15,728,640 waypoints and corresponds to 1,310,720 hours of maritime navigation data, providing a substantial foundation for model training. The model received the data with windows sliding within different MMSI codes. The train, validation, and test datasets were split by MMSI code and are free of data leakage.

### E.3. Stage 2 Database Sampling Strategy

To sample the database for open-set verification, the dataset is partitioned into multiple voyage segments based on Maritime Mobile Service Identity (MMSI) and Key Points. Within each segment, we apply a sliding-window approach to extract trajectory sequences, with a window size of $L_{seq}$ and a stride of $S$ steps, where $L_{seq}$ denotes the sequence length for model input. In practice, we set it as 288, i.e., 24 hours.

After the sliding-window operation, we perform stratified sampling by randomly selecting 50 trajectories per key node, achieving a good balance between representation diversity and computational efficiency. Key Points with insufficient samples (fewer than 50 points) are excluded from the final dataset. This methodology yields a balanced dataset comprising 56 ports and 2,800 data entries, ensuring adequate representation across different geographical regions.

### E.4. Data Description and Coverage for Public Dataset

We further evaluate our method on a public AIS dataset published on Hugging Face to assess generalization beyond our private data source. Each AIS record contains the vessel identifier (MMSI), timestamp (BaseDateTime), geolocation (latitude and longitude), kinematic attributes (speed over ground, course over ground, and heading), vessel type encoded using standard AIS ship-type codes, navigation status, and a track identifier when available.

The version used in our experiments exhibits broad spatial and kinematic coverage.

- Latitude range: $0.05° N$ to $61.25° N$

- Longitude range: $177.86° W$ to $171.22° E$

- SOG has a mean of $11.98 \pm 4.39$ knots with the range $[1.00, 29.90]$

- COG has a mean of $185.73 \pm 101.72$ with the range cover the full angular domain $[0°, 360°)$

Overall, the dataset predominantly covers near-equatorial to mid- and high-latitude regions in the Northern Hemisphere and includes trajectories across major ocean basins. It is qualified as a test dataset.

The public dataset contains multiple vessel categories, including Cargo, Fishing, Passenger, Pleasure Craft/Sailing, Tanker, and Tug/Tow. To ensure consistency with our private dataset and experimental focus, we restrict evaluation to cargo vessels by selecting AIS messages whose VesselType falls within the cargo ship-type codes (70–79).

According to the repository statistics, the dataset comprises AIS messages from approximately 19k vessels with hundreds of millions of records. In the snapshot used for evaluation, we observe 19,016 unique MMSIs, with individual vessel trajectories spanning from several days to multiple months, supporting both short- and long-horizon prediction scenarios.

To align preprocessing with our private dataset, we apply a unified pipeline: AIS messages are filtered by vessel type, grouped by MMSI (and TrackID when available), and resampled via linear interpolation to a fixed temporal resolution of $T = 5$ minutes. All features are cast to consistent units and data types to ensure compatibility across datasets. The process details are the same as the private dataset. The private test dataset has 2324 samples, while the public test dataset has 560 samples and comes from distinct sources.

# F. Implementation Details

### F.1. Model Implementation

The hidden size of MiniMind Block was set as 256 with 2 key-value heads. MiniMind Model 1 has 1 hidden layer, while MiniMind Model 2 has 1, and the hidden size of the MLP is 128.

### F.2. Training Details

The learning rate is 7e-5. We also utilized AdamW as the optimizer and Cosine Annealing Warm Restarts as the scheduler. Other hyperparameters are default. $t$ in Evaluation was set as 15. All training and evaluation are performed on a single NVIDIA RTX 4090.

