# OpenReview forum: "SKETCH: Semantic Key-Point Conditioning for Long-Horizon Vessel Trajectory Prediction"
_ICML.cc/2026/Conference — ICML 2026 regular_

### Official Review · Reviewer_x7G2 · 2026-03-12

**Soundness:** 3
**Presentation:** 2
**Significance:** 3
**Originality:** 3
**Overall Recommendation:** 4
**Confidence:** 2

**Summary:**

This paper addresses the problems of trajectory drift and global direction inconsistency in long-horizon vessel trajectory prediction by proposing a hierarchical modeling framework named SKETCH. The framework introduces the concept of a Next Key Point (NKP) as a semantic intention variable, decomposing the trajectory prediction task into two stages: global semantic decision-making (predicting the NKP) and local motion modeling (conditional prediction based on the NKP).

A retrieval- and contrastive-learning–based verification mechanism is adopted. By matching historical trajectories with those in a reference database, the method determines the most semantically consistent target point through a voting mechanism, thereby overcoming the limitations of closed-set classification in handling unseen key points.

The framework employs MiniMind (a Transformer variant) as the decoder. Given historical trajectories and the predicted NKP, it autoregressively generates future increments of SOG (Speed Over Ground) and COG (Course Over Ground), which are finally mapped to latitude–longitude coordinates. Experiments on a real AIS dataset show that SKETCH improves long-horizon prediction accuracy (e.g., 24-hour forecasts) by 40–60% compared with existing state-of-the-art (SOTA) models such as TrAISformer, while also significantly improving the global shape and curvature consistency of predicted trajectories.

**Compliance With Llm Reviewing Policy:**

Affirmed.

**Key Questions For Authors:**

For new routes with no historical reference trajectories, or in remote ocean regions where the reference database is extremely sparse, how does the model maintain the reliability of NKP prediction?

NKP is defined as an “equivalence class sharing the same navigational semantics.” Could the authors provide more details on how these semantics are automatically annotated during preprocessing, particularly in complex port areas?

**Strengths And Weaknesses:**

Strengths

By introducing the semantic NKP, the framework successfully decouples high-level navigational intent from low-level physical motion, effectively constraining the prediction space of future trajectories and mitigating uncertainty accumulation over long distances.

The retrieval-based NKP prediction method (Retrieval-based Verification) enables the model to handle geographical locations not explicitly defined in the dataset, improving robustness in dynamic maritime environments.

Weaknesses

The inference process heavily depends on the scale and quality of the reference trajectory database. If a shipping route lacks relevant cases in the database, the NKP voting mechanism may fail or produce misleading guidance.

The framework strongly relies on the accuracy of the first-stage NKP prediction.

The paper lacks additional hyperparameter sensitivity analyses, such as experiments on the cosine similarity threshold.

---

> ### Author Rebuttal · Authors · 2026-03-31
>
> We sincerely thank you for your thorough review and valuable comments and appreciate the opportunity to explain certain aspects of our work.
> Firstly, regarding the concern about database sparsity and unseen routes, we would like to clarify that we have evaluated our method on a public dataset that is not used in either training or the reference database, and the model still achieves strong performance. This suggests that the proposed framework is robust to database sparsity. From an industrial perspective, even if performance degrades in regions with limited coverage, the framework remains highly extensible: additional trajectories from those regions can be directly incorporated into the database without retraining, along with their corresponding NKP annotations. This is a key advantage of our open-set design, which allows scalable improvement as more data becomes available.
> Secondly, regarding hyperparameter sensitivity, we have conducted additional experiments on cosine similarity thresholds, as provided in the supplementary material (https://anonymous.4open.science/r/Rebuttal_anonymous-FBFF/). In practice, we used the top one based on similarity rather than voting in the original experiments, and we tested different thresholds for cosine similarity reported in the above link. The top-one policy achieves the best.
> Thirdly, regarding the definition and annotation of NKP semantics, we adopt a spatial rule-based approach during preprocessing. Specifically, straits and ports are defined as geographic lines. If a trajectory intersects with the line, the corresponding segment is assigned that strait or point as its NKP, capturing the shared navigational semantics of passing through that region. This design groups trajectories with similar navigational intent into equivalence classes, enabling consistent semantic supervision even in complex maritime environments.

---

> > ### Author Rebuttal · Reviewer_x7G2 · 2026-04-04
> >
> > Thank you for your rebuttal, I will maintain my score.

---

> > > ### Author Response · Authors · 2026-04-08
> > >
> > > Thank you very much for your thorough follow-up and for confirming that our response has adequately addressed your concerns. We sincerely appreciate your time, constructive feedback, and thoughtful evaluation throughout the review process. Your comments helped us improve the clarity and rigor of the paper. Please feel free to let us know if you need further clarification.

---

### Official Review · Reviewer_pgbZ · 2026-03-12

**Soundness:** 3
**Presentation:** 3
**Significance:** 2
**Originality:** 2
**Overall Recommendation:** 4
**Confidence:** 3

**Summary:**

This paper studies long-horizon vessel trajectory prediction and argues that existing methods fail partly because they do not explicitly represent high-level navigational intent. To address this, the paper introduces a hierarchical formulation in which future trajectories are predicted by conditioning on a latent Next Key Point (NKP), and reformulate it into a hierarchical forecasting problem. The method uses a decoder-only Transformer backbone, a contrastive retrieval-style NKP predictor, and a multi-stage training procedure that separates motion modeling from semantic intent estimation. The empirical section reports improvements over MP-LSTM and TrAISformer on both a private AIS test set and a public AIS dataset, and also includes ablations on NKP quality and qualitative visualizations.

**Compliance With Llm Reviewing Policy:**

Affirmed.

**Final Justification:**

The response has adequately addressed my main concerns. In particular, the clarification of the differences between NKP and prior intent-aware approaches helps clarify the novelty of the proposed formulation; the newly added comparisons with stronger recent baselines strengthen the empirical support for the method; and the additional horizon-wise analysis better supports the paper’s claimed advantages in the long-horizon setting. Overall, the rebuttal has improved my confidence in the technical contribution and evaluation

**Key Questions For Authors:**

1. How does the proposed NKP formulation differ from prior intent-, goal-, or destination-aware methods such as SEMINT, MENet, and what are its main advantages over them?

2. Can the authors compare their method against a broader set of stronger and more recent baselines to better support the claimed state-of-the-art performance?

3. Can the authors provide additional long-horizon experiments, such as horizon-wise evaluations, to more directly demonstrate the method’s advantage as the prediction range increases?

**Limitations:**

The method is only validated through offline evaluation on historical AIS data, so the risks of real-world maritime deployment remain unclear. The authors should also discuss the method’s dependence on NKP inference and the reference database, as well as its sensitivity to incorrect NKP predictions. A clearer discussion of these practical limitations would better clarify the paper’s practical applicability.

**Strengths And Weaknesses:**

Strengths
One of the strength of the paper is its problem reformulation. Treating long-horizon forecasting as a hierarchical problem with an explicit semantic intent variable is a sensible and well-motivated idea for maritime prediction, where route-level decisions and local control naturally operate at different levels. The NKP formulation is conceptually interesting because it aims to constrain future rollouts to semantically feasible subsets rather than relying only on low-level kinematics.

A second strength is the overall architectural design, which is coherent and well aligned with the paper’s hierarchical formulation. The proposed framework integrates retrieval-based NKP verification, a three-stage learning strategy, and an inference-time conditioning pipeline in a logically consistent way. In particular, the use of retrieval and voting over a reference database is a sensible choice for supporting open-set generalization beyond a purely closed-set classifier.

The experiments are encouraging. On the reported private test set, the method improves over both baselines on MSEP, MSEC, MFD, and inference time; on the public AIS dataset, it again achieves the best reported numbers and shows a large gap over TrAISformer. The ablations on NKP quality are also useful because they show that correct or well-predicted NKPs matter for downstream trajectory smoothness and global consistency.

Weaknesses
1. About the paper’s positioning and motivation.
   The core motivation of the paper—improving long-horizon vessel trajectory prediction through explicit intent-, goal-, or destination-level modeling—appears closely related to prior works such as “SEMINT”, “MENet”. However, the paper does not provide sufficient comparison or discussion with these motivation-levelly similar methods. As a result, it remains unclear how the proposed NKP formulation differs from these approaches and what its specific advantages are, which weakens the paper’s ability to clearly highlight its novelty and contribution.

2. Limited comparison with strong baselines.
   While “MP-LSTM” and “TrAISformer” are relevant references, a stronger case for state-of-the-art long-horizon forecasting would require evaluation against a broader set of recent and stronger methods, such as graph-based models, continuous-state approaches (“AIS-ACNet”), and newer transformer- or generative-based predictors (“AISFuser”, “DiffuTraj”, “STGDPM”, “SEMINT”). Without such comparisons, it is difficult to determine whether the reported gains mainly come from the proposed NKP formulation or from the relatively narrow baseline selection.

3. Limited experiments of the method in the long-horizon setting.
   Since the paper is explicitly motivated by long-range forecasting, it would be helpful to include additional experiments that more directly show how the proposed method behaves as the prediction horizon increases. For example, the authors could provide “horizon-wise evaluations” or stratified results over different forecast ranges, similar to recent long-term vessel trajectory prediction studies such as “SEMINT”. Such experiments would better support the paper’s central motivation and make its claimed long-horizon advantage more convincing.

[1] Chen, N., Yang, A., Wu, H., Chen, L., Xiong, W., & Jing, N. (2025). SEMINT: an LLM-empowered long-term vessel trajectory prediction framework. International Journal of Geographical Information Science, 39(9), 1938–1972.

[2] Zhang, X., Liu, J., Chen, C., Wei, L., Wu, Z., & Dai, W. (2025). Goal-driven long-term marine vessel trajectory prediction with a memory-enhanced network. Expert Systems with Applications, 263, 125715.

[3] Shin, Y., Kim, N., Lee, H., In, S. Y., Hansen, M., & Yoon, Y. (2024). Deep learning framework for vessel trajectory prediction using auxiliary tasks and convolutional networks. Engineering Applications of Artificial Intelligence, 132, 107936.

[4] Zhang, Z., Yuan, W., Fan, Z., Song, X., & Shibasaki, R. (2025). AISFuser: Encoding Maritime Graphical Representations With Temporal Attribute Modeling for Vessel Trajectory Prediction. IEEE Transactions on Knowledge and Data Engineering, 37(4), 1571–1584.

[5] Li, C., Gan, Y., Lan, T., Cai, Y., Liu, X., Lin, R., & Liu, Q. (2024). DiffuTraj: A Stochastic Vessel Trajectory Prediction Approach via Guided Diffusion Process. arXiv preprint arXiv:2410.09550.

[6] Jin, W., Tang, H., & Zhang, X. (2025). STGDPM: Vessel Trajectory Prediction with Spatio-Temporal Graph Diffusion Probabilistic Model. In Database Systems for Advanced Applications (DASFAA 2025, Part II), 571–586.

---

> ### Author Rebuttal · Authors · 2026-03-31
>
> We sincerely thank you for your thorough review and valuable comments, and we appreciate the opportunity to clarify certain aspects of our work.
> Firstly, regarding the concern about the relation to prior intent-aware methods such as SEMINT and MENet, we would like to clarify that NKP is not limited to destination level semantics (e.g., ports), but also includes important maritime structures such as straits like Strait of Hormuz and key transit regions like Suez Canal, since trajectories in these regions are complex. This provides richer and more informative intent compared to destination-only modeling. More importantly, from a modeling perspective, NKP fundamentally changes how the trajectory prediction problem is formulated. Unlike destination-only models that predict a single distant target, NKP automatically decomposes the long-horizon trajectory into several shorter-horizon and provides intermediate hierarchical waypoints that guide the decoder’s low-level motion prediction segment by segment, making the prediction problem more practical and continuously feasible. In contrast, destinations are often far from the current position and may not provide as efficient information as NKP. We acknowledge that this distinction was not sufficiently emphasized and will clarify it in the revised manuscript.
> Besides, regarding the concern on comparisons with stronger baselines, we have conducted additional experiments on several recent methods, and the results are provided in the supplementary material (https://anonymous.4open.science/r/Rebuttal_anonymous-FBFF/), where our method remains competitive among all reproduced baselines. Due to time and resource limitations, we were only able to reimplement a subset of them and will further extend these comparisons in the final version. In particular, some methods are not directly applicable under our setting: AIS-ACNet is designed for multi-vessel interaction modeling and requires multiple vessels (typically ≥2–3 ships) within the same temporal window to construct graph structures, while our dataset consists of globally distributed trajectories with sparse temporal overlap, which does not satisfy this requirement. SEMINT relies on an external LLM-based chatbot to infer high-level intent, introducing dependencies on external systems and prompting, which makes the pipeline difficult to reproduce and control. Moreover, it is trained under its own official setup, which differs from our data pipeline. Due to these differences and computational constraints, we evaluated it in a cross-dataset manner as reported in the supplementary material.
> Finally, regarding the long-horizon evaluation, we agree that this is important for supporting our motivation. To directly address this concern, we have conducted additional horizon-wise experiments, which are also included in the supplementary material (link above). The results show that our method maintains more stable performance as the prediction horizon increases, and we will incorporate this analysis more explicitly in the revised manuscript.

---

> > ### Author Rebuttal · Reviewer_pgbZ · 2026-04-08
> >
> > Thank you for your detailed rebuttal and for the additional experimental evidence. Your response has adequately addressed my main concerns. In particular, the clarification of the differences between NKP and prior intent-aware approaches helps clarify the novelty of the proposed formulation; the newly added comparisons with stronger recent baselines strengthen the empirical support for the method; and the additional horizon-wise analysis better supports the paper’s claimed advantages in the long-horizon setting. Overall, the rebuttal has improved my confidence in the technical contribution and evaluation, and I will raise my score.

---

> > > ### Author Response · Authors · 2026-04-08
> > >
> > > Thank you very much for your careful re-evaluation and for updating your score after considering our response. We are truly grateful that you found our additional clarifications and experiments sufficient to resolve your concerns. Your feedback was very valuable in helping us strengthen both the technical presentation and the framing of our contributions. Please feel free to let us know if you need further clarification.

---

### Official Review · Reviewer_fzFB · 2026-03-13

**Soundness:** 2
**Presentation:** 2
**Significance:** 2
**Originality:** 2
**Overall Recommendation:** 3
**Confidence:** 4

**Summary:**

This paper studies long-horizon vessel trajectory prediction from AIS data and argues that existing sequence models fail to preserve global directional consistency over long horizons. The main idea is to introduce a high-level latent variable called the Next Key Point (NKP), intended to represent navigational intent, and to factor trajectory prediction into two components: predicting the NKP from history and then predicting the future trajectory conditioned on both history and NKP. The method uses a multi-stage pipeline: first train a conditional trajectory predictor with oracle NKP supervision, then freeze the backbone and train an NKP module using a contrastive/retrieval-style objective, and finally perform inference by predicting NKP and conditioning the trajectory predictor on it. The paper also proposes a coordinate update based on SOG/COG and evaluates the approach on one private AIS dataset and one public AIS dataset, reporting gains over MP-LSTM and TrAISformer in positional, curvature, Frechet, and runtime metrics.

**Compliance With Llm Reviewing Policy:**

Affirmed.

**Key Questions For Authors:**

1. What exactly is the evaluation protocol for the public dataset? The paper states that no model is trained or fine-tuned on that dataset, but the inference section also states that the reference database contains training trajectories for both the public and private test datasets. Please clarify whether any public-dataset trajectories are used in the retrieval database at test time. If the answer is no, my confidence in the generalization claim would increase. If the answer is yes, the claim should be reframed more carefully.

2. How much of the improvement comes from the hierarchical factorization versus access to explicit NKP supervision derived from predefined spatial annotations? A stronger ablation would compare against baselines given the same extra supervision, or against alternative conditioning signals, to isolate the benefit of the proposed factorization.

3. Why are the reported MSEP values in Table 5 nearly identical across several very different NKP settings, including correct NKP, wrong NKP, and the 4-channel baseline? Please explain whether this is due to rounding, horizon averaging, or some other effect, and ideally provide standard deviations over multiple runs.

4. Why is MSEC reported as 0.00 for MP-LSTM in Table 2 if the text says MSEC does not apply to MP-LSTM because the method predicts only two points and interpolates the rest? Please clarify whether 0.00 means “not applicable,” “undefined but displayed as zero,” or an actual computed value.

5. Can the authors compare against stronger recent baselines, especially methods cited in the paper such as TrackGPT or AISFuser, or explain why such comparisons are not feasible?

**Limitations:**

No. The paper should more explicitly discuss at least: dependence on manually defined key nodes and NKP annotations; the fact that the main dataset is proprietary; the restriction to cargo vessels in the public evaluation; and possible failure modes when the inferred NKP is wrong or when a route does not align well with predefined ports/straits.

**Strengths And Weaknesses:**

The paper addresses a relevant forecasting problem, and the high-level intuition is reasonable: explicitly modeling intent may help stabilize long-horizon predictions. I also appreciate that the authors do not only report pointwise position error, but also add curve-level and smoothness-oriented metrics, plus runtime, which gives a broader view of behavior. The ablation around correct versus wrong NKP is also useful, since it helps show that the conditioning signal is not ignored by the model. The private-data split is stated to be by MMSI, which is the right direction for leakage control at the vessel level.

That said, I have several concerns about soundness. First, the main conceptual contribution is not fully isolated experimentally. The trajectory model is trained with oracle NKP labels, where NKPs are obtained from predefined spatial annotations such as ports and straits, yielding 103 annotations. This makes the setting closer to supervised waypoint conditioning than to discovering a genuinely latent semantic variable. As written, it remains unclear how much of the gain comes from the hierarchical factorization itself versus simply injecting additional structured supervision unavailable to the baselines.

Second, the evidence for the open-set claim is not fully convincing. The paper emphasizes generalization to unseen key points, but the stage-2 database construction keeps only key points with at least 50 samples and ends with 56 ports / 2,800 entries, while Table 4 reports accuracy for settings that appear to mix closed-set and open-set variants. More importantly, the inference section says that the reference database contains training trajectories “for both the public and private test datasets,” while the public-dataset experiment also states that none of the models are trained or fine-tuned on the public dataset. The exact protocol here is hard to parse: if retrieval uses public-dataset trajectories, even without gradient updates, that is materially different from strict out-of-domain generalization and should be described much more carefully.

Third, the baseline set is too narrow for an ICML paper. The quantitative comparison is only against MP-LSTM and TrAISformer, while the paper cites more recent work such as TrackGPT and AISFuser but does not compare against them. Since the proposed contribution is partly about long-horizon robustness and generalization, stronger modern baselines would be important. Without them, it is difficult to judge whether the empirical gains reflect a real advance over the current state of the art or mainly over a limited comparison set.

Fourth, some quantitative results raise questions. In Table 5, MSEP is reported as 0.41 for correct NKP, SFT-O, SFT-C, wrong NKP, and pure 4ch, while MSEC and MFD vary substantially; pretrain-C even has lower MSEP but much worse MFD. This pattern is not impossible, but it is unusual enough that it deserves a clearer explanation, confidence intervals, or repeated runs. Likewise, the text says MSEC does not apply to MP-LSTM because intermediate points are interpolated, yet Table 2 still reports 0.00 for MP-LSTM. That makes the metric comparison somewhat ambiguous.

Fifth, the theoretical and physical-analysis sections do not add much support in their current form. Appendix A proves generic facts about conditioning reducing uncertainty, but those results are standard and do not validate the specific modeling assumptions used here. Similarly, the coordinate-update section and stability appendix claim improved numerical stability relative to spherical formulations, but the empirical check is only a one-step displacement consistency test with error around 10^-9, which mainly shows the update formula is internally consistent, not that it improves forecasting quality. There is no direct ablation against a standard geodesic/spherical update inside the same model.

On presentation, the paper is readable at a high level, but it needs polishing. There are multiple grammar issues and some notation/statement mismatches, for example “Under Theorem 3.2” when the paper earlier gives Assumption 3.2, and several passages where the wording is imprecise. Some implementation details are also too sparse for full reproducibility: hidden sizes, similarity threshold, learning rate, optimizer, and scheduler are listed, but many other hyperparameters are left as defaults, and the main dataset is proprietary.

---

> ### Author Rebuttal · Authors · 2026-03-31
>
> We sincerely thank you for your thorough review and valuable comments, and we appreciate the opportunity to explain certain aspects of our work.
> Firstly, regarding the concern that the gain may come from explicit supervision rather than hierarchical factorization, we would like to clarify that the word "hierarchical" in our paper corresponds to high-level navigational intent, while low-level module refers to dynamical motion at each time step. We aim not only to predict the trajectory but also the next key points. Therefore, we explicitly supervised the NKP. However, we derived the latent expression in the second stage.
> Secondly, regarding the concern about the public dataset protocol, after careful verification we confirm that the reference database does not contain any trajectories from the public dataset. The statement in the inference section was a typo, and Section 5.2 is correct: none of the evaluated models are trained or fine-tuned on the public dataset, and all results reflect true out-of-domain generalization. We will revise the manuscript to remove this ambiguity.
> Thirdly, regarding the limited baselines, we have conducted additional comparisons with more recent methods, as provided in the supplementary material (https://anonymous.4open.science/r/Rebuttal_anonymous-FBFF/), where our method remains competitive among all reproduced baselines. Due to time and resource constraints, we were only able to reimplement a subset, and we will further extend these comparisons in the final version.
> Fourthly, regarding the unusual metric behavior, we agree that this requires clarification. We verified the results across multiple runs and confirmed their consistency. The key reason is that MSEP measures local point-wise error, while MFD evaluates global trajectory shape and curvature, which is more aligned with long-horizon prediction quality. Therefore, different NKP settings may yield similar local errors but significantly different global trajectory structures. We will add variance statistics and further explanation in the revised manuscript. For MSEC, since MP-LSTM predicts sparse points and interpolates intermediate steps, the resulting trajectory is definitely smooth, making MSEC not meaningful in this case; the reported value of 0.00 reflects this degeneracy rather than a comparable metric.
> Fifthly, regarding the physical formulation, we observed that directly predicting latitude/longitude with Transformer-based models often leads to instability and accumulated bias over long horizons, whereas predicting speed (SOG) and direction (COG) and updating coordinates via physical equations improves numerical stability and robustness. The analysis in Appendix A verifies the correctness of the formulation; while it is not intended as a theoretical novelty, it addresses a practical limitation that the distance between the consecutive points predicted by Transformer-based architecture gradually narrows. If needed, we can further provide ablation comparisons with direct coordinate prediction in the revision.
> Finally, we thank the reviewer for pointing out issues in grammar and presentation. We will carefully revise the manuscript for clarity and consistency. Our implementation is based on PyTorch, and while some hyperparameters (e.g., beta for Adam) follow standard defaults, we will provide a more complete specification for reproducibility. Code, model checkpoints, and datasets will be fully released upon acceptance.

---

> > ### Author Rebuttal · Reviewer_fzFB · 2026-04-03
> >
> > Thank you for the detailed rebuttal and the additional supplementary comparisons. The extra results are helpful and address part of my concern about the originally narrow baseline set. In particular, the added comparisons against additional methods and the horizon-wise MFD analysis strengthen the empirical case that the proposed approach is beneficial for long-horizon prediction. However, I still do not think my main concerns are fully resolved. The central remaining issue is whether the gains primarily come from the proposed hierarchical NKP factorization itself or from access to explicit NKP supervision derived from predefined spatial annotations. This is important to the paper’s claimed contribution, and I still think stronger controlled ablations would be needed to isolate that effect more convincingly.

---

> > > ### Author Response · Authors · 2026-04-07
> > >
> > > Thank you for the thoughtful follow-up. We found that the observed gain arises from both the proposed hierarchical NKP factorization and access to explicit NKP supervision derived from predefined spatial annotations. To address this more directly, we added stronger controlled ablations that separately isolate these two effects.
> > >
> > > Specifically, we compare four variants under the same evaluation protocol:
> > >
> > > - **M00 (Pure 4ch baseline):** The model takes only the standard four-channel history (lat/lon/sog/cog) as input and predicts future trajectories directly, without any NKP supervision or NKP-based factorized conditioning. **Its MFD is 8.99.**
> > > - **M01 (factorization without NKP supervision):** We retain the factorized architecture by using the MLP output as an intermediate conditioning representation for trajectory generation, but remove all supervision derived from ground-truth NKP annotations; this intermediate representation is learned solely through the trajectory prediction loss. **Its MFD is 7.93.**
> > > - **M10 (NKP supervision without factorized use):** The model still takes only the standard four-channel history as input, and Encoder 1 / MiniMind Model 1 are trained with NKP-aware supervision through the MLP branch; however, the predicted NKP representation is not explicitly used as an intermediate conditioning variable for trajectory generation. **Its MFD is 7.87.**
> > > - **M11 (full model):** The model uses both explicit NKP supervision and factorized NKP-conditioned trajectory generation, i.e., it first predicts NKP-related information from the four-channel history and then explicitly uses it as an intermediate variable to guide trajectory prediction. **Its MFD is 7.78.**
> > >
> > > These results suggest the following. First, **factorization** itself contributes: comparing M01 against M00, introducing a factorized intermediate variable already improves MFD from 8.99 to 7.93, even without any explicit NKP supervision. Second, **explicit NKP supervision** also contributes: comparing M10 against M00, NKP-aware supervision alone improves performance from 8.99 to 7.87, even when the trajectory branch does not explicitly use NKP as an intermediate conditioning variable. Third, **the best result is obtained when both are combined**: M11 further improves to 7.78, outperforming both M01 and M10. This indicates that the final gain is not explained by supervision alone or factorization alone, but by their **combination**.
> > >
> > > So, we would like to clarify that our intended claim is not that all gains are attributable solely to hierarchical factorization nor solely to supervision. Rather, the proposed framework benefits from both:
> > >
> > >  (i) structured NKP supervision derived from predefined spatial annotations, and
> > >
> > >  (ii) explicit factorized use of NKP as an intermediate variable for trajectory generation.
> > >
> > >  The new controlled ablations shown above are intended to clarify this attribution and to show that the two components are complementary rather than redundant.

---

### Decision · Program_Chairs · 2026-04-30

**Decision:**

Accept (regular)

**Comment:**

The paper proposes a hierarchical framework for long-horizon vessel trajectory prediction using a latent variable, the Next Key Point (NKP), to capture navigational intent. All reviewers find the paper well-motivated, with pgbZ and x7G2 noting the formulation and design as strengths. Several concerns are raised by the reviews, which are addressed by the author rebuttal. fzFB questions whether the gains are derived from the proposed modeling or from supervision, which are addressed by the results from additional ablations. pgbZ notes limited experimentation while x7G2 requires sensitivity analysis, with both satisfied by the author rebuttal. Overall, the ACs agree that the paper proposed a novel and well-written approach to an interesting problem with sufficient validation, so recommend acceptance at ICML. The authors are encouraged to incorporate all reviewer feedback in the final version.